# Border-Zone Infarction Due to Cerebrovascular Fibromuscular Dysplasia

**DOI:** 10.3390/diagnostics12061337

**Published:** 2022-05-27

**Authors:** Yu-Ming Chen

**Affiliations:** 1Department of Neurology, Hualien Tzu Chi Hospital, Buddhist Tzu Chi Medical Foundation, Hualien 97002, Taiwan; yumingchenmd@gmail.com; Tel.: +886-919-848213; 2School of Medicine, Tzu Chi University, Hualien 97004, Taiwan

**Keywords:** cerebrovascular fibromuscular dysplasia, string-of-beads sign, ischemic stroke in young adults

## Abstract

A 45-year-old male presented with acute-onset left-sided weakness and slurred speech. Non-contrast-enhanced brain magnetic resonance imaging revealed cortical and internal border-zone infarcts compatible with stroke. A survey of ischemic stroke risk factors in young adults excluded coagulopathy, vasculitis, and cardiac disease. Nevertheless, neck-computed tomography angiography revealed a long-segmental narrowing of the right internal carotid artery with wall thickening and a “string-of-beads” appearance suspicious for fibromuscular dysplasia, which was confirmed on further angiography. His clinical condition stabilized after intensive medical therapy. This case demonstrates cerebrovascular fibromuscular dysplasia as a possible cause of ischemic stroke in young adults.


Figure 1A 45-year-old male presented with acute-onset left-sided weakness and slurred speech. The clinical diagnosis of stroke was based on focal neurologic deficits, persistent symptoms, and an abrupt clinical course. Non-contrast brain magnetic resonance images revealed border-zone infarcts (Figure 1a). Magnetic resonance angiography (MRA) disclosed trivial flow in the right internal carotid artery (Figure 1b). A survey of the ischemic stroke risk factors in young adults excluded coagulopathy, vasculitis, hematologic and cardiac disease. A history of heavy smoking (two packs a day for the last 27 years), hypertension, and dyslipidemia (low-density lipoprotein: 143 mg/dL) were noted during stroke workup. Nevertheless, neck computed tomography angiography (CTA) revealed a long-segmental narrowing of the right internal carotid artery with wall thickening and a “string-of-beads” appearance suspicious for fibromuscular dysplasia (FMD), which was confirmed on further angiography (Figure 2). Adequate hydration, aspirin, and statin were prescribed for secondary stroke prevention. His clinical condition stabilized after intensive medical therapy. (**a**). On diffusion-weighted images, the cortical and internal border-zone ischemic infarcts are visualized with hyperintense lesions. (**b**). On magnetic resonance angiography image, the flow signal intensity of the right proximal internal carotid artery is severely reduced with trivial blood flow (white arrowhead). Complete occlusion of the right distal internal carotid artery and left middle cerebral artery are also noted.
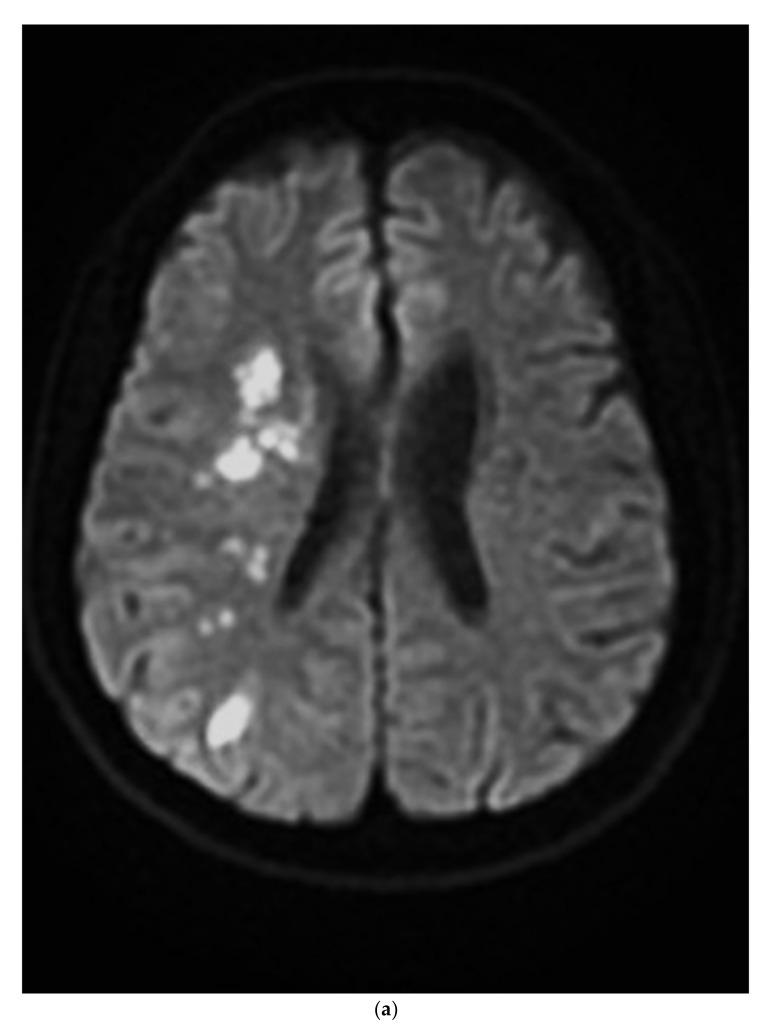

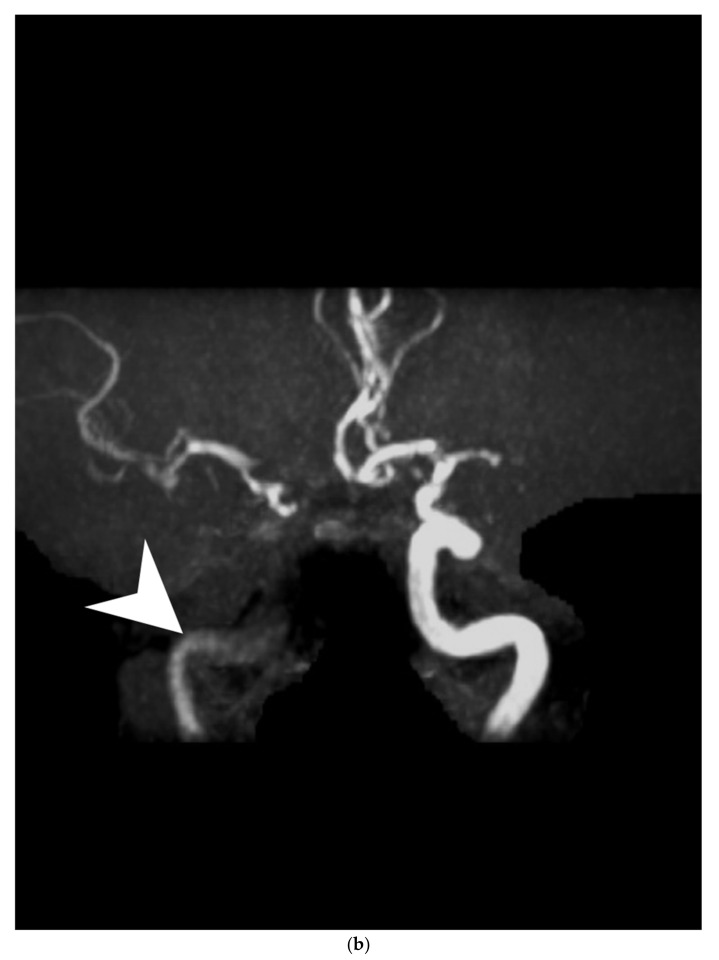

Figure 2FMD is characterized by noninflammatory, nonatherosclerotic vasculopathy with no identifiable underlying cause [1]. FMD predominantly involves small to median-sized arteries and affects middle-aged individuals (30 to 50 years old) [2]. According to recent systemic review, cerebrovascular FMD is as common as renal FMD [2]. In contrast to sites susceptible to the atherosclerotic process (proximal internal carotid artery), cerebrovascular FMD commonly involves the middle to distal portion of the internal carotid artery. The most common symptoms of cerebrovascular FMD are pulsatile tinnitus and headache [3]. However, cerebrovascular FMD is frequently asymptomatic and is found by accident through an image examination. Devastating neurologic consequences, such as stroke and transient ischemic attack, can occur in symptomatic patients. Diagnosis of cerebrovascular FMD requires non-invasive and invasive image exams, including CTA, MRA and digital subtraction angiography (DSA). Non-invasive CTA and MRA are the modalities of choice whereas DSA remains the gold standard for the diagnosis of cerebrovascular FMD. Nevertheless, because of its invasiveness and risk of arterial dissection of the fragile vessels, DSA is commonly reserved for ambiguous cases and those requiring endovascular treatment. The typical imaging findings of cerebrovascular FMD are alternating luminal narrowing and dilatation, resulting in a “string-of-beads” appearance. Fusiform vascular ectasia and vascular loop are also typical features [4]. Other less typical features include aneurysm, arterial dissection and subarachnoid hemorrhage [4]. The stroke mechanisms of cerebrovascular FMD are likely heterogeneous, such as cerebral hypoperfusion, cardioembolism, and artery-to-artery embolism [2]. In this case, cerebrovascular FMD resulted in hypoperfusion of the right internal carotid artery, and further caused the right cortical and internal border-zone ischemic infarction. The stroke mechanism is supported by long-segmental severe stenosis of the right internal carotid artery. Subsequent DSA disclosed a classic “string-of-beads” sign in the cervical internal carotid artery (Figure 2). There was no evidence of cardiac emboli or artery-to-artery emboli after 24 h Holter monitoring, echocardiogram, or carotid ultrasound (without atherosclerotic plaque but reduced flow velocity in the right internal carotid artery). The management principles of stroke with cerebrovascular FMD due to arterial stenosis are comparable to stroke without cerebrovascular FMD [5]. During the acute phase of an ischemic stroke, intravenous thrombolysis and endovascular therapy are recommended in eligible patients [6]. This patient exceeded the time window for intravenous thrombolysis and endovascular therapy. Thus, an antithrombotic agent with aspirin was prescribed during the acute phase. For the long-term management of secondary stroke prevention, medical therapy should be customized according to stroke mechanisms and co-morbidities for optimal outcomes [7]. Long-term aspirin, antihypertensive agents, statin, and smoking cessation were prescribed for this patient. It is worth mentioning that endovascular therapy, such as carotid stenting or surgical arterial bypass, is typically reserved for patients with recurrent ischemic events despite optimal medical therapy [6,8,9]. At the 18-month follow-up, the patient’s neurologic deficits recover gradually with minimal left hand dexterity impairment. He also manages his chronic diseases well with optimal medication and lifestyle modification (i.e., smoking cessation / increased physical activity). Of particular importance is the relatively young age of the patient, which is the reason for the comprehensive survey of stroke etiologies. In young patients with ischemic stroke, secondary stroke prevention can be properly achieved only when the true stroke etiology is found. To sum up, we demonstrated cerebrovascular FMD as a possible cause of ischemic stroke in young adults. This differential diagnosis should be considered when stroke occurs at a young age with a typical “string-of-beads” sign on CTA, MRA, or DSA. Digital subtraction angiography shows long-segmental alternating luminal narrowing and dilatation of the right internal carotid artery with a typical “string-of-beads” sign (white arrowheads). The diagnosis of cerebrovascular fibromuscular dysplasia is confirmed.
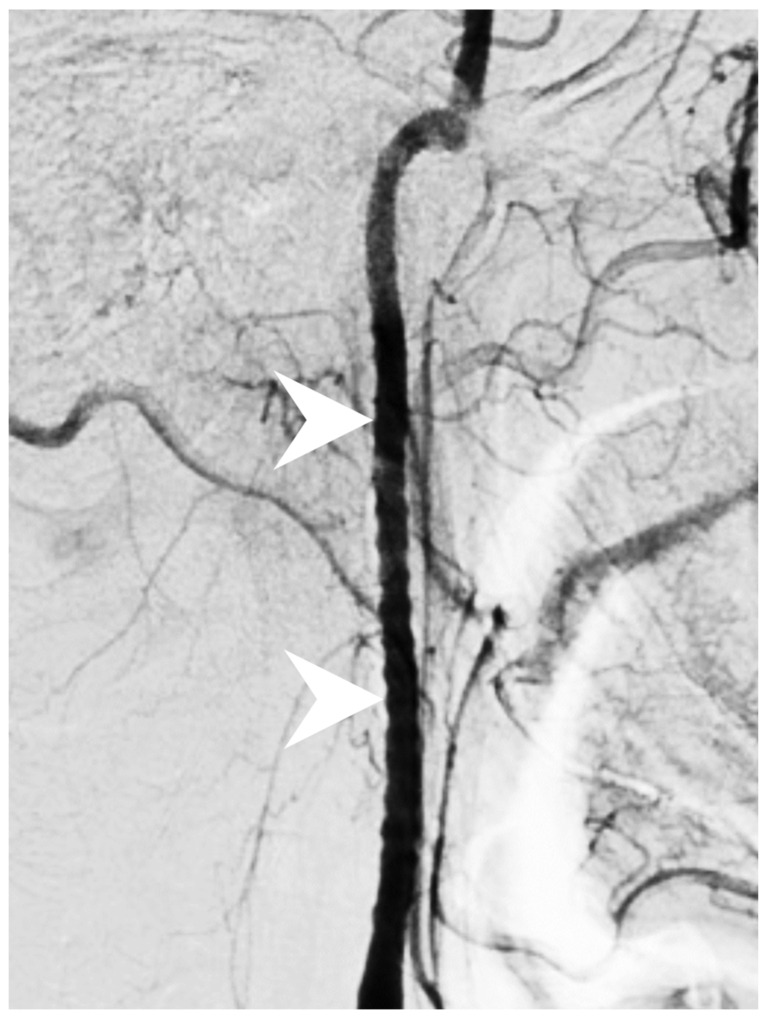



## Data Availability

Not applicable.

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
