# Peer review of "Border-Zone Infarction Due to Cerebrovascular Fibromuscular Dysplasia"

_diagnostics, 2022, doi:10.3390/diagnostics12061337_

Round 1
Reviewer 1 Report
This article (case report) highlights that a possible cerebral fibromuscular dysplasia coud be as a significant cause of stroke in young adults.
References should be updated and implemented.
Image quality should be improved.
Furthermore, figures legends should be made more exhaustive.
The English language and style are fine but a minor spell check is required.
Reviewer 2 Report
The content of this paper must be increased especially in the conclusions and in discussion. It seems necessary to find morte ideas of originality in the paper and more bibliograpical references.
Reviewer 3 Report
Wrong use of abbreviation for digital subtraction angiography, not DXA but DSA. Wrong use of English term such as "through"; wrong style of writing such as "24-hour Holter " not "24 hour Holter"; and syntax error such as "Cardiac emboli or artery-to-artery emboli mechanisms is less likely after 24 hour Holter monitoring, echo-cardiogram and carotid ultrasound (without atherosclerotic plaque but reduced flow velocity in the right internal carotid artery).". Since this is a case report, it may not be correct for the authors to conclude that "we emphasize cerebral fibromuscular dysplasia as a significant cause of stroke in young adults" rather than possible diagnosis of ischemic stroke in young adults. There were inadequate discussion regarding the medical therapy in this patient and related literature review and the findings in follow-up, both clinical and neuroimaging. First onset of stroke in 45 years is not considered as young stroke. (Typically defined as those with first onset of stroke at the age below 45 years old) There were very little information regarding this patient, include co-morbidity, risk factors and life-styles. All these information are important when decide for medical therapy. Options other than medical therapy endovascular therapy of surgical bypass procedure should also be discussed.
Round 2
Reviewer 2 Report
none
Reviewer 3 Report
The authors have addressed all the comments from the previous review.